# Application of W-Doped VO₂ Phase Transition Mechanism and Improvement of Hydrophobic Self-Cleaning Properties to Smart Windows

Xiaoxian Song [1,2], Ze Xu [1,2], Dongdong Wei [1,2], Xuejie Yue [3], Tao Zhang [3], Haiting Zhang [1,2,*], Jingjing Zhang [1,2], Zijie Dai [1,2] and Jianquan Yao [1,2,4]

[1] School of Mechanical Engineering, Jiangsu University, Zhenjiang 212013, China; songxiaoxian@ujs.edu.cn (X.S.); 2222103022@stmail.ujs.edu.cn (Z.X.); 2222103154@stmail.ujs.edu.cn (D.W.); 1000005146@ujs.edu.cn (J.Z.); daizijie@ujs.edu.cn (Z.D.); jqyao@tju.edu.cn (J.Y.)

[2] Institute of Micro-Nano Optoelectronics and Terahertz Technology, Jiangsu University, Zhenjiang 212013, China

[3] School of Chemistry and Chemical Engineering, Jiangsu University, Zhenjiang 212013, China; yuexuejiechem@163.com (X.Y.); zhangtaochem@163.com (T.Z.)

[4] School of Precision Instruments and Opto-Electronics Engineering, Tianjin University, Tianjin 300072, China

* Correspondence: zhanghaiting@ujs.edu.cn

**Abstract:** A passive responsive smart window is an emerging energy-saving building facility that does not require an active energy supply due to its passive excitation characteristics, which can fundamentally reduce energy consumption. Therefore, achieving passive excitation is the key to the application of such smart windows. In this paper, VO₂ is used as a critical raw material for the preparation of smart windows, and we researched the feasibility of its phase transition function and hydrophobic self-cleaning function. VO₂ has the characteristic of undergoing a reversible phase transition between metal and insulator under certain temperature conditions and can selectively absorb spectrum at different wavelengths while still maintaining a certain visible light transmission rate, making it a reliable material for smart window applications. The one-step hydrothermal method was used in this work, and different concentrations of tungsten (W) elements were utilized for doping to reduce the VO₂ phase transition temperature to 35 °C and even below, thus adapting to the ambient outdoor temperature of the building and enabling the smart window to achieve a combined solar modulation capability of 14.5%. To ensure the environmental adaptability and anti-fouling self-cleaning function of the smart window, as well as to extend the usage period of the smart window, we have modified the smart window material to be hydrophobic, resulting in an environmental surface contact angle of 152.93°, which is a significant hydrophobic improvement over the hydrophilic properties of inorganic glass itself. The realization of the ideal phase transition function and the self-cleaning function echoes the social trend of environmental protection, enriches the use of scenarios and achieves energy saving and emission reduction.

**Keywords:** smart windows; phase transition; tungsten doping; solar modulation capability; hydrophobic self-cleaning

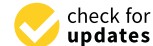



## 1. Introduction

The operation of society and the productive order of life require a large amount of energy. However, the building sector accounts for one-third of the global energy mix and about 15% of CO₂ emissions to maintain a livable room temperature [1]. Doors and windows are more likely to lose heat than walls. Also, they are the main areas of energy consumption in buildings [2]. Compressors are commonly used for cooling and controlling temperature in most buildings and houses, which often emit large amounts of greenhouse gases, leading to an increasingly serious greenhouse effect. A smart window is a scientific product for the energy-saving and low-carbon environment. Reasonable development and

application can respond to the national government's energy-saving and environmental protection policy.

VO$_2$ is a very interesting material that can undergo phase transition with temperature changes [3]. Heated to a certain temperature, it can change from a monoclinic phase to a rutile phase, which brings out its semiconducting properties. This phenomenon is known as the phase transition principle [4,5]. Therefore, more experiments and studies are needed to gain insight into the principles and factors—affecting the phase transition of VO$_2$. The phase transition is influenced by lots of factors. VO$_2$ undergoes a phase transition from a monoclinic phase to a rutile phase as it rises to a certain temperature, which indicates that temperature is the most vital factor. The rutile phase has the properties of electrically low resistance and low transmission to infrared. On the contrary, with decreased phase transition temperature, VO$_2$ is inverted from a rutile phase back to a monoclinic phase, which has the properties of electrically high resistance and high transmission to infrared [6]. The optical behavior of smart windows before and after the phase transition is visible and translucent [7,8]. In addition, defects and impurities in the VO$_2$ material and the crystal structure, grain size and sparseness of VO$_2$ can reflect on the phase transition process [9–11]. It is crucial for the investigation and improvement of VO$_2$ material applications, but their phase transition temperatures are either high or low, resulting in immense limitations in their applications. Therefore, we need to target the selection of materials with suitable phase transition temperatures for relevant smart windows research. Up to now, VO$_2$ has become the first choice for building materials such as smart windows due to the phase transition temperature (68 °C). The heat of solar radiation is mainly concentrated in the near-infrared band [12,13], and can be blocked by VO$_2$ in high-temperature environments [14]. After the temperature is reduced, VO$_2$ will continue to undergo phase transition so that the light in the near-infrared band with radiant heat can be transmitted. During the heating and cooling cycle above, the transmittance in the visible band remains almost unchanged, thus maintaining high visible-light transmittance (T$_{lum}$) and excellent solar modulation ability (ΔT$_{sol}$) [15,16]. This is a thermochromic smart window that automatically responds to atmospheric temperature, regulates room temperature and ensures visible light transmission without additional energy supply [8]. The introduction of phase transition materials into buildings is a revolutionary development in the field of architecture.

The phase transition temperature of VO$_2$, 68 °C, is almost close to the outdoor sunlight temperature in the tropics and subtropics, and the doping of elements can reduce the phase transition temperature in order to make the smart windows more diversified [9,10]. Using different vanadium sources and reducing agents as hydrothermal reaction systems, VO$_2$ with different morphologies can be prepared, regulating different reaction times and varying the filling efficiency of the reactor [17]. The common product morphologies are mainly granular and rod-like. VO$_2$ nanorods show essential metallic and insulated phase transitions in the longitudinal direction. They possess fast response time and high sensitivity, which makes them suitable for devices such as sensors and switches [18–20], while VO$_2$ nanoparticles are commonly used in thermal control, catalytic reactions and energy storage devices [21–23]. Shen et al. prepared VO$_2$ nanoparticles using a hydrothermal method and employed Joule heating to contribute the temperature to the phase transition conditions of VO$_2$ [24]. He et al. prepared W-doped VO$_2$ nanorods and synthesized hydrogel thin film smart windows [25], and Zhang et al. tried five ratios of tungsten (W) doping and compared the performance of VO$_2$ smart windows using in situ testing [26]. These studies on smart windows used different temperature settings to change the phase transition conditions of VO$_2$, which greatly reflected the consequence of different doping on the phase transition function, but the usage scenario is relatively single. So, this paper emphasized not only its application characteristics as a building material but also the development of self-cleaning properties. In addition, due to the hysteresis of the VO$_2$ phase transition, the smart windows maintain low transmission in the NIR band for a considerable period of time during the temperature cooling process, which also greatly improves the blocking performance and autonomous regulation of high-temperature IR by the smart

windows [24,27]. The adaptability of thermochromic smart windows to complex external environments makes them an effective choice as an energy-efficient building material. The contact angle between the smart window and water is generally 30–40°, so the glass can easily form water droplets, which do not fall off easily. During the evaporation and drying process, it is liable to adsorb the dust and dirt on the glass in the environment by long-term accumulation, which directly affects the thermal regulation ability of the smart window. For the reasons above, we need to research and discuss the addition of its automatic cleaning function [28]. The surface self-cleaning function of building materials is generally divided into two kinds: superhydrophilic self-cleaning and superhydrophobic self-cleaning. Both kinds can achieve self-cleaning and anti-fouling; only the anti-fouling principle is different [29,30]. Hydrophilic self-cleaning uses ultraviolet light to oxidize organic compounds so that the glass has a super hydrophilic surface [31,32]; however, due to the high surface hardness of glass, the hydrophilic material is easy to fall off. In contrast, hydrophobic self-cleaning is a technique similar to the 'lotus leaf effect' [33], where water droplets falling on a superhydrophobic surface with a contact angle greater than 150° are carried away in the process of rolling out of the glass surface through its papillary nanostructure, ensuring that the glass is dry and permeable. Zhou et al. used a two-step wet chemical graphical layer technique to prepare a robust oleophobic hydrophobic fabric [34]. Simultaneously, the quick-drying surface formed by superhydrophobic self-cleaning can also provide effective protection to the phase transition coating of the smart window and extend the service life of the phase transition material in the self-cleaning application of thermochromic smart windows.

In this paper, three sets of W-$VO_2$ nanoparticles containing different doping were prepared by hydrothermal method to investigate the phase transition temperature and variation of $VO_2$ smart windows. At a doping concentration of 1.7 at. % of tungsten, the rising phase transition temperature of $VO_2$ dropped to 33.51 °C, with a near-infrared transmittance of 31% and a solar modulation capability of 9.8%, reflecting the excellent infrared blocking ability as well as thermal modulation capability of $VO_2$, which can be applied as a smart window material [35–37]. It used hexafluoropropene (PVDF-HFP) as the main raw material for the preparation of hydrophobic coatings as well, with contact angles reaching superhydrophobic standards [29], and proved its excellent self-cleaning function in water-drop decontamination experiments.

## 2. Experimental Section

### 2.1. Chemicals and Materials

Vanadium (IV) sulfate oxide hydrate ($VOSO_4 \times H_2O$); ammonium metatungstate hydrate (($NH4)_6H_2W_{12}O_{40} \times H_2O$); polyvinylpyrrolidone PVP10, K29-32; and poly (vinylidene fluoride-co-hexafluoropropylene) drugs were purchased from McLean Biochemical Technology Co., Ltd., Shanghai, China. Tetraethylorthosilicateammonium hydroxide; -dimethylformamide; and hydrazine hydrate 85% were purchased from China National Pharmaceutical Group Chemical Reagent Co., Ltd., Shanghai, China.

### 2.2. Fabrication of W-$VO_2$ Nanoparticles

Three sets of W-$VO_2$ nanoparticles, labeled W1 (0.0 at. %), W2 (1.7 at. %), and W3 (2.0 at. %), were prepared for this experiment. For the preparation of the W2 sample, 2.0 g $VOSO_4$ and 0.0517 g $H_{28}N_6O_{41}W_{12}$ (0 g for W1 and 0.0609 g for W3) were dissolved in 80 mL of deionized water to obtain a blue solution and then 0.8 mL $N_2H_4$ (85 wt. % aqueous solution) was added dropwise until the solution turned grey and stirring was continued for 60 min. The solution was adjusted to a pH of 9 by adding 0.15 M NaOH dropwise with constant stirring. The grey precipitate was then vacuum-filtered and washed with deionized water. The precipitate was dispersed in 80 mL of deionized water and continuously sonicated for 2 h to form a slurry. The slurry was then transferred to a 100 mL reaction vessel and hydrothermally heated at 240 °C in a blast oven for 36 h. After natural cooling to room temperature, the black precipitate was vacuum-filtered and washed several

times with deionized water and ethanol, respectively, and finally dried at 60 °C for 10 h under vacuum.

### 2.3. Fabrication of VO$_2$ Film on the Inorganic Glass

The VO$_2$ nanofilms were prepared by sonication using a typical dispersion system of 2 wt. % W-VO$_2$ and 5 wt. % PVP-K30. The prepared W-VO$_2$ powder was then mixed with PVP-K30 and deionized water, stirred for 60 min and sonicated for 10 h. The suspension was uniformly poured onto the glass and dried at 90 °C for 5 min.

### 2.4. Fabrication of Superhydrophobic Modification

A 1.5 wt. % solution of silica nanoparticles was first prepared. Firstly, a small amount of ammonium hydroxide and 50 mL of ethanol were mixed to form a homogeneous solution, then tetraethyl orthosilicate and silicone resin were added and magnetically stirred for 3 h to form a hydrophobic silica particle sol. Next, 1.0 g PVDF-HFP was mixed in 50 mL DMF to form a homogeneous PVDF-HFP solution. After 1 h of magnetic stirring, the solution was ready for coating. In addition, using a two-step coating method, the soaked silica nanoparticle solution and the PVDF-HFP solution were applied sequentially, then dried at 130 °C for 1 h and the above steps were repeated multiple times.

### 2.5. Material Characterization

The morphology of the synthesized W-VO$_2$ nanoparticles was characterized using scanning electron microscopy (SEM, QUANTA 250 FEG, Thermo Fisher Scientific, Shanghai, China). The crystalline phase of W-VO$_2$ was obtained by X-ray diffraction (XRD, D8 Advance, Bruker, Billerica, MA, USA). The surface chemistry and elemental composition of the prepared W-VO$_2$ nanoparticles were verified by X-ray photoelectron spectroscopy (XPS, AXIS ULTRA DLD, Shimadzu, Suzhou, China). The transmittance spectra were measured at different temperatures in the UV, visible (300–1100 nm) and NIR (1100–2000 nm) bands using a transmission emission test system (7EMSpec, Saifan, Beijing, China) equipped with a home-made precision thermostatic console (KER 4100-08S), where the visible and UV bands were received by a UV-enhanced silicon detector (7ID219) and the NIR band by an indium gallium arsenic detector (7ID3233). The phase transition properties of VO$_2$ were measured using differential scanning calorimetry (DSC, NETZSCH DSC 200F3, NETZSCH, Shanghai, China) in a nitrogen gas stream at temperatures ranging from 10 to 80 °C with an elevation rate of 3 °C per minute. The phase transition temperature and latent heat values were obtained using the analysis software supplied with the thermogravimetric analyzer. The contact angle of the hydrophobic smart window was measured using a video contact angle meter (JY-82C, Dingsheng, Chengde, China), measuring range: 0–180°. For the three sets of samples in this experiment, the following Equations (1) and (2) were used to calculate the luminous transmittance (T$_{lum}$) (400–760 nm), the near-infrared transmittance (T$_{NIR}$) (1100–2000 nm) and the solar transmittance (T$_{sol}$) (300–2000 nm), respectively, based on the measured transmission spectra.

$$T_i = \frac{\int \varphi_i(\lambda) T(\lambda) d\lambda}{\int \varphi_i(\lambda) d\lambda} \tag{1}$$

$$\Delta T_{sol} = T_{sol}(M) - T_{sol}(R) \tag{2}$$

In the formula, $T(\lambda)$ is the transmittance at wavelength $\lambda$; $i$ refers to "lum", "NIR", or "sol"; $T$ denotes temperature and Tc denotes phase transition temperature; $\Delta T_{sol}$ denotes solar modulation capacity; $\varphi_{lum}$ is the standard luminous efficiency function of light vision; $\varphi_{sol}$ is the solar irradiance spectrum of air mass 1.5 (corresponding to the sun at 37° above the horizon); (M) and (R) represent the two different phases of VO$_2$, respectively.

## 3. Results and Discussion

### 3.1. Hydrothermal Mechanism and Morphology Observation

In this work, a hydrothermal synthesis method was adopted to prepare $VO_2$ with different doping (see experimental section for details), the process is shown in Figure 1. The hydrothermal synthesis method has mild reaction conditions, which includes high-temperature and high-pressure reactors, and uses water as the reaction medium to dissolve substances with little or no solubility under normal temperature and pressure in a high-temperature and high-pressure reaction-friendly environment [38]. It is suitable for large-scale production because of low preparation costs, a stable crystalline phase, controlled morphology and so on [39]. During the preparation of $VO_2$ nanoparticles, the filling rate of the reactor was roughly 80%, and nitrogen was added to the kettle to prevent residual oxygen in the kettle from joining the oxidation reaction. $VO_2$ nanoparticles with different doping amounts were characterized by SEM (Figure 2).

The SEMs of zero doped, 1.7 at. % W-doped and 2.0 at. % W-doped $VO_2$ nanoparticles are shown in Figure 2a–c, respectively. Figure 2d(i,ii) represents the TEMs of typical $VO_2$ nanoparticles with different sizes. As it is demonstrated that there is no obvious distinction between pure $VO_2$ particles and the two other doping, the sample consists of spherical particles of hundreds of nanometers. One-step hydrothermal is a nucleation-growth-transformation process. The doping of W leads to an increase in nucleation sites that result in a more uniform distribution and size of $VO_2$ particles. The formation of different morphologies of $VO_2$ is related to the direction of growth and transformation, which may be splitting, exfoliation, recrystallization, or regrowth [40,41]. In Figure 2e, we can see that the film thickness of the $VO_2$ smart window without hydrophobic coating is roughly 1 micrometer, and Figure 2f shows that the film thickness with a hydrophobic coating is roughly 2 μm. The thickness of $VO_2$ coating is approximately the same as that of hydrophobic coating.

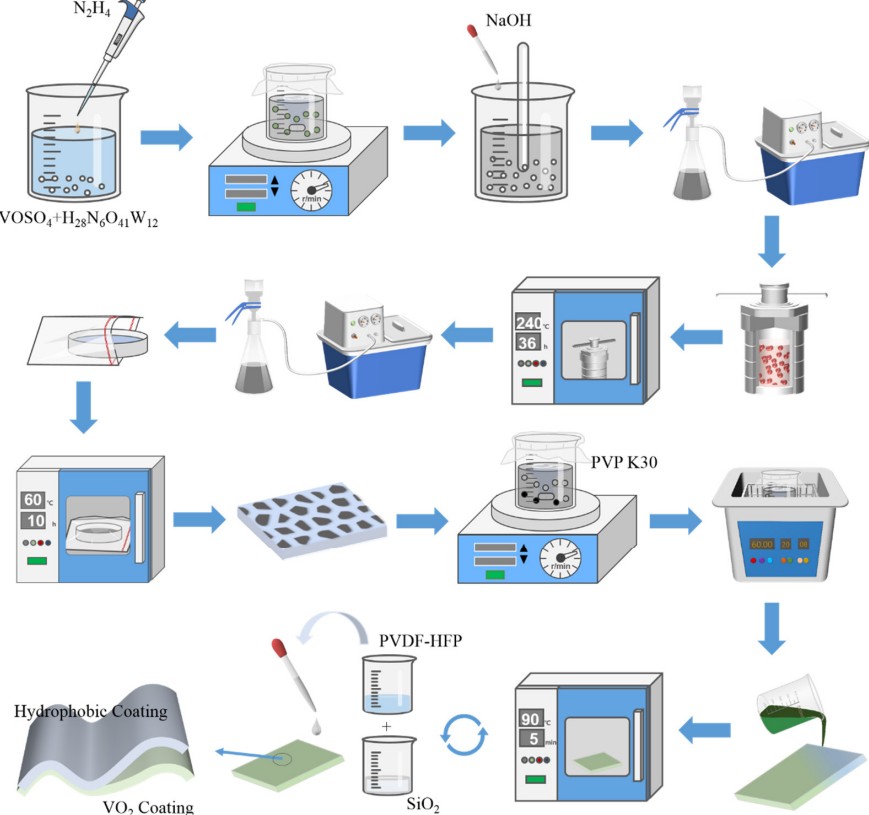

**Figure 1.** Schematic illustration of the synthesis of W-$VO_2$ and the fabrication process of hydrophobic films.

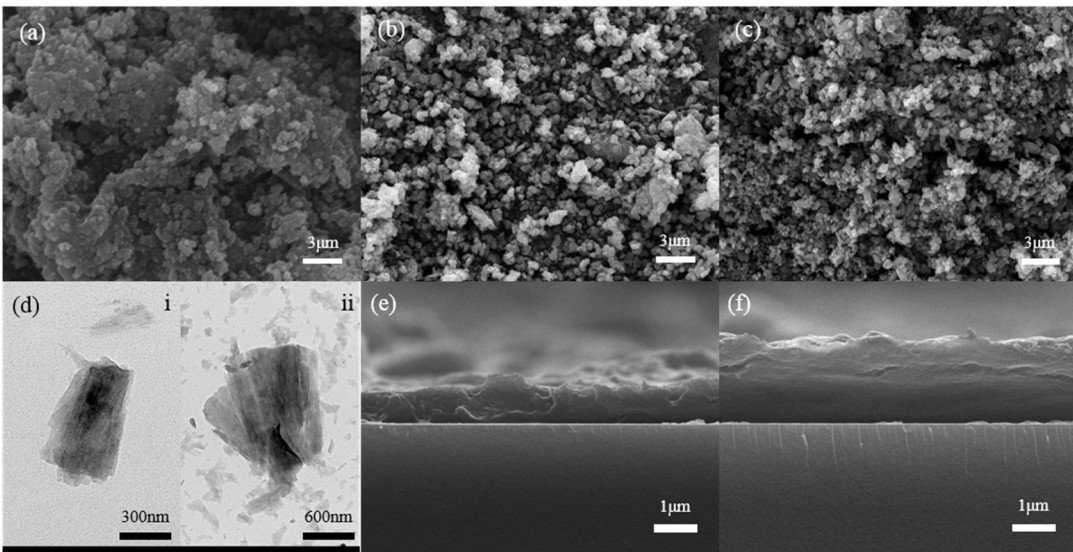

**Figure 2.** (**a**–**c**) The SEM images of three different gradient W-doped VO₂ nanoparticles. (**d**) The TEM images of 1.7 at. % W-doped VO₂ nanoparticles, i and ii represent the morphology at different magnifications. (**e**,**f**) The SEM images of VO₂ smart windows' cross-sectional with and without hydrophobic coating.

Moreover, further performance analysis of three sets of VO₂ nanoparticles with different doping concentrations was carried out in this work, and their XRD patterns show clear diffraction peaks (Figure 3a), indicating good crystallinity. These peaks could be assigned to monoclinic VO₂ (M) (JCPDS Card No. 44-0525), and no ammonium metatungstate or other w-containing oxides were detected, indicating good doping. The diffraction peak shifted from 27.92 to 27.76 as the doping concentration increased (Figure 3b). This peak shift suggests that the interplanar distance in the (011) plane extends from 3.19 to 3.22 Å due to the incorporation of larger W atoms in place of smaller V atoms.

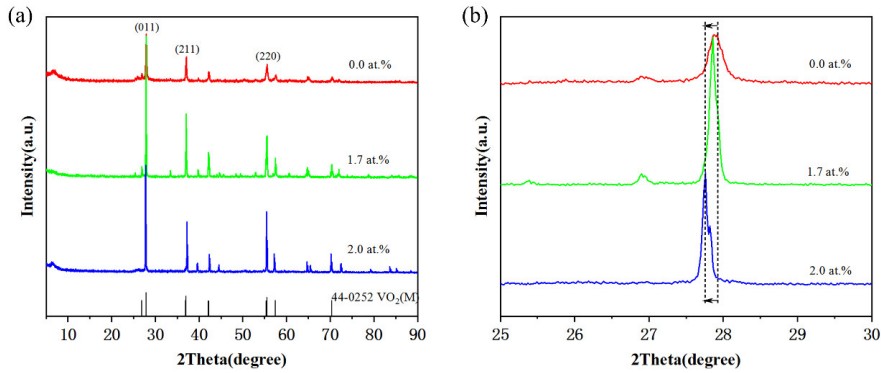

**Figure 3.** The crystal structure of VO₂ nanoparticles with different W contents. (**a**) The XRD patterns of the as-synthesized VO₂ powder with different concentrations of W dopants. (**b**) Magnifications of the smoothed XRD pattern depicted in the $25° \leq 2\theta \leq 30°$ range.

XPS was detected on W-doped 1.7 at. % VO₂ nanoparticle powders (Figure 4) and four elements, W, O, V and C, were detected in the XPS spectra (Figure 4a). The presence of element C was due to adsorption contamination (Figure 4b). Analyzed after performing element C calibration, the high-resolution XPS spectra clearly showed that the W-doped samples had W4f₅/₂ at 37.73 eV and W4f₇/₂ at 35.73 eV (Figure 4d) [42]. The vanadium peaks V3p, V2p₁/₂ and V2p₃/₂, as well as the oxygen peak O1s, are also clearly distinguished in the high-resolution spectra (Figure 4c,d). By fitting the high-resolution XPS spectra of the VO₂ sample doped with W at 1.7 at. % (Figure 4e,f), the main nuclear en-

ergy level peaks at 35.53 eV and 37.73 eV for $W4f_{7/2}$ and $W4f_{5/2}$. This indicates that the hydrazine hydrate only undergoes reduction and does not react further. Both $V^{4+}$ and $V^{5+}$ were present in the sample [43]. $V2p_{3/2}$ and $V2p_{1/2}$ peaks can be deconvoluted into two peaks: 517.73 eV and 516.53 eV have binding energies pointing to $V2p_{3/2}$ and 525.03 eV and 523.13 eV have binding energies pointing to $V2p_{1/2}$. As mentioned previously, the $V2p_{3/2}$ peak at 516.53 eV indicates the +4 valence state of vanadium, while 517.73 eV indicates the +5 valence state of vanadium [44]. This is due to the oxidation of the particle surface.

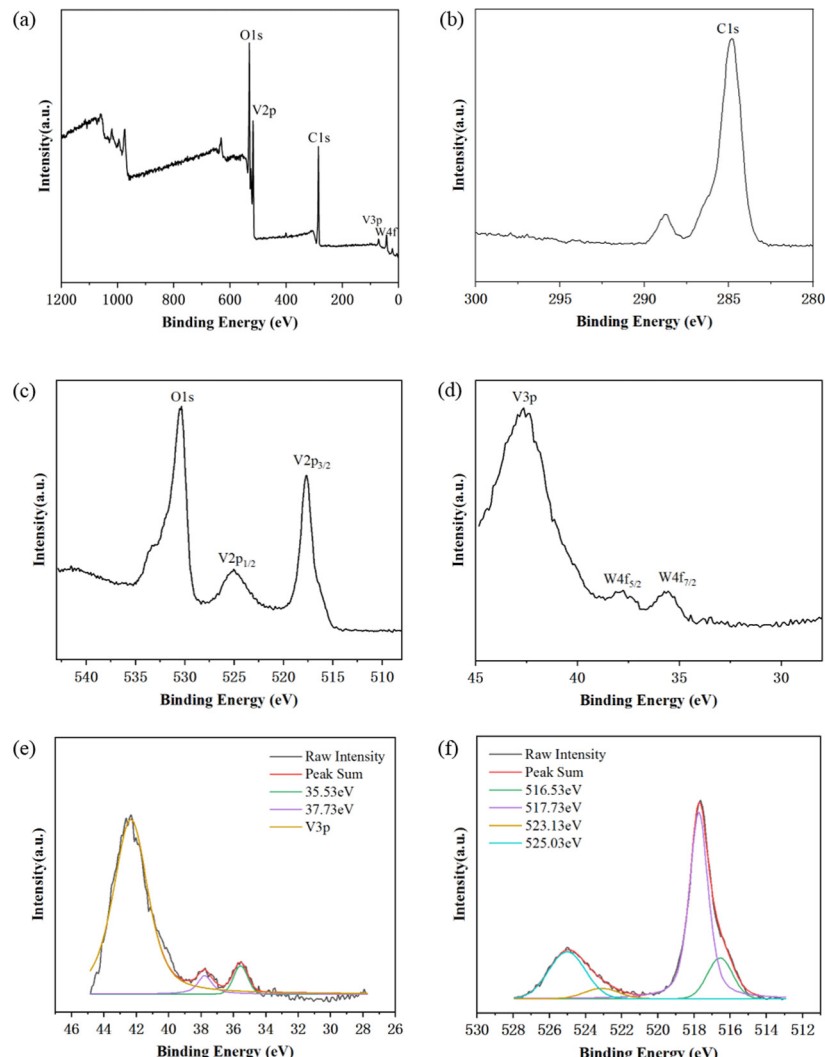

**Figure 4.** XPS spectra of W-VO$_2$ sample (1.7 at. %). (**a**) XPS survey spectra. (**b**) High-resolution spectra for C 1s peak. (**c**) High-resolution spectra for O 1s and V 2p peaks. (**d**) High-resolution spectra for V 3p and W 4f peaks. (**e**,**f**) Fitting to high-resolution W 4f and V 2p peaks of the sample.

### 3.2. Thermal Regulation Performance of Smart Windows

The energy-saving performance of VO$_2$ smart windows depends, to a large extent, on their phase transition temperature (T$_c$). The performance characterization has demonstrated that doping can change the phase transition temperature of VO$_2$. It is known from energy band theory that the VO$_2$ phase transition is caused by a change in the position relationship between the orbitals as the temperature changes, causing the electron motion to change from continuous to discontinuous, thus showing the properties of a conductor and a semiconductor [45].

As shown in Table 1, different doping elements have different effects on the phase transition temperature. Dopant ions are generally chosen as cations with larger ionic radii than $V^{4+}$ and higher valence, such as $W^{6+}$, $Mo^{6+}$, $Nb^{5+}$ or anions with larger ionic radii than

$O^{2-}$ [46,47], such as $F^-$. Conversely, the introduction of small radii and low valence states, such as $Al^{3+}$, $Cr^{3+}$, $Ga^{3+}$ and $Ge^{4+}$, would lead to higher phase transition temperatures [48]. On balance, the element W was selected for doping in this study, and the phase transition temperature was measured specifically using differential scanning calorimetry.

**Table 1.** The effect of doping of different elements in $VO_2$ on its phase transition temperature [49].

| Element (at.%) | W | F | Nb | Mo | Cr | Ge | Ga | Al |
|---|---|---|---|---|---|---|---|---|
| $T_c$ variation (°C) | −23 | −20 | −7.8 | −7.5 | +3 | +5 | +6.5 | +9 |

As shown in Figure 5a, the DSC curves of three different doped $VO_2$, combined with Table 2, the phase transition temperature of 0-doped $VO_2$ is 67.46 °C during heating, which is also almost consistent with the 68 °C reported in the literature [24], the doping of W element very effectively reduces the phase transition temperature, the 1.7 at. % doping reduces the phase transition temperature at a rate of −19.97 °C/at. % during heating, while the 2.0 at. % doping reaches −22.96 °C/at. %, which shows that the amount of doping also greatly affects the phase transition rate of $VO_2$. The higher the amount of doping, the faster the rate of change of phase transition temperature.

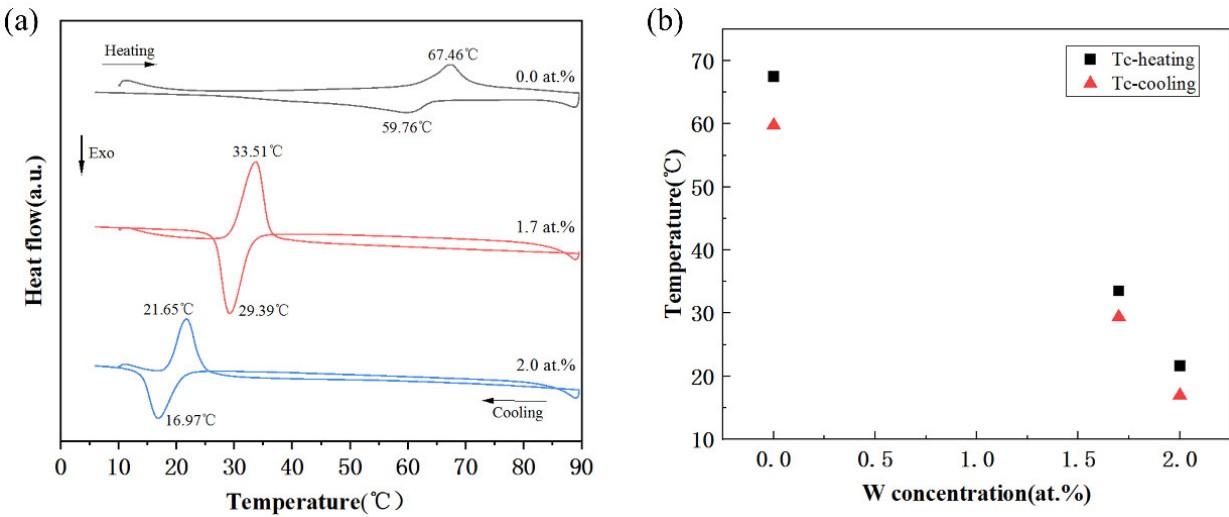

**Figure 5.** The phase transition properties of $VO_2$ nanoparticles with different W contents. (**a**) DSC curves of W-$VO_2$ powders. (**b**) The phase transition temperatures of W-doped $VO_2$ powders.

**Table 2.** Summary of phase transition temperatures during the heating and cooling cycles of $VO_2$ (M) samples prepared with different W-doped concentrations.

| W-Doped Ratio (at.%) | $T_{c\text{-heating}}$ (°C) | Latent Heat (J/g) | $T_{c\text{-cooling}}$ (°C) | Latent Heat (J/g) | Hysteresis Width (°C) | $T_c$ |
|---|---|---|---|---|---|---|
| 0 | 67.46 | 18.13 | 59.76 | 14.03 | 7.7 | 63.61 |
| 1.7 | 33.51 | 32.63 | 29.39 | 33.17 | 4.12 | 31.45 |
| 2.0 | 21.65 | 20.68 | 16.97 | 22.77 | 4.68 | 19.31 |

During the cooling cycle, a reversible phase transition from $VO_2$ (R) to $VO_2$ (M) can be observed to continue, and the phase transition temperature during the cooling process decreases, resulting in a hysteresis width, which can clearly be seen to decrease as the amount of doping increases.

The spectral transmittance of the prepared $VO_2$ nanoparticle films is shown in Figure 6a,b. In Figure 6a, the transmittance of the $VO_2$ nanoparticle films with different doping amounts in the wavelength range of 300–1100 nm is similar at different temperatures, while in the near-infrared wavelength range of 1100–2000 nm (Figure 6b), as the

doping amount increases and the ambient temperature reaches 90 °C, the transmittance of VO$_2$ thin films significantly decreases. The above is also consistent with the theoretical phase transition characteristics of VO$_2$. Figure 6c shows the temperature dependence of the transmittance of the 1.7 at. % doped VO$_2$ smart window at 1800 nm (T$_{1800}$). It can be seen that the smart window can still maintain a low transmittance during the cooling down to 35 °C due to the presence of hysteresis, which is also consistent with the hysteresis width of the previous DSC curve. The variation curves of luminous transmittance T$_{lum}$ and solar modulation efficiency ΔT$_{sol}$ with W doping concentration are shown in Figure 6d.

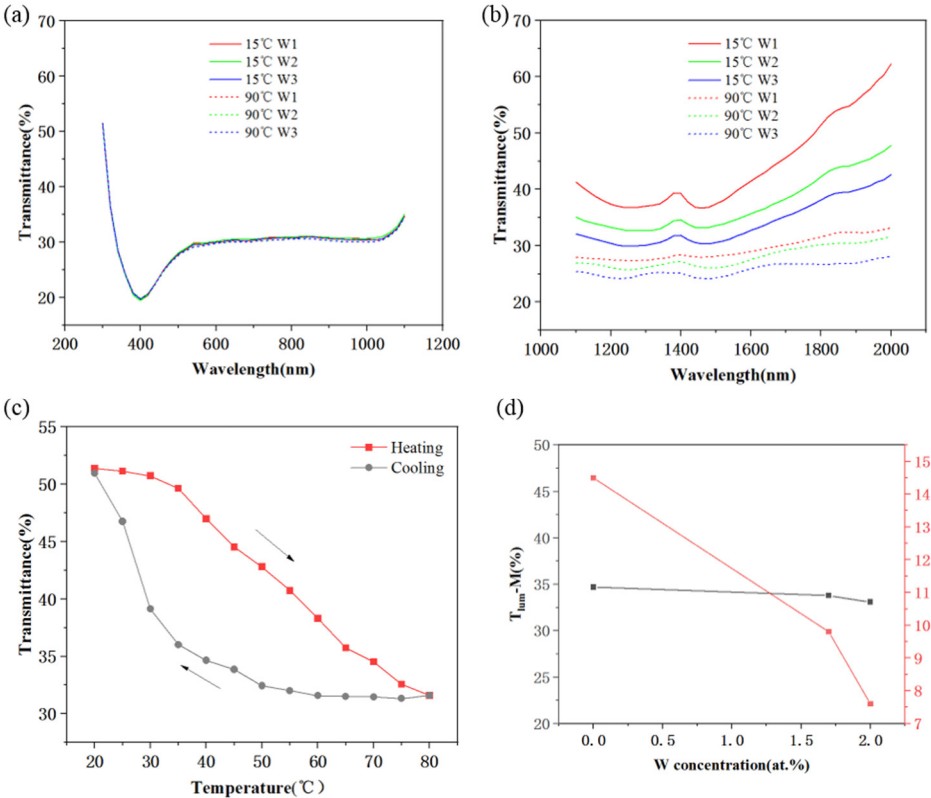

**Figure 6.** The thermochromic property of W-doped VO$_2$ nanoparticle films. (**a**,**b**) The transmittance spectra of VO$_2$ films with different W contents (in different colors) at room temperature (solid lines) and 90 °C (dash lines). (**c**) The $T_{1800}$−temperature hysteresis loops of W-VO$_2$ films. (**d**) Plots of $T_{lum}$-M and $\Delta T_{sol}$ as a function of W-doped concentration.

### 3.3. Hydrophobic Performance and Self-Cleaning Inspection

In practice, there are many surfaces that also encounter various kinds of dirt, mud, dust and their mixtures, and in the face of this, nano hydrophobic self-cleaning coatings are able to cope. In order to improve the hydrophobic self-cleaning function of glass-based smart windows and to ensure their light transmission rate, a two-step coating method is employed in this paper, with the PVDF-HFP solution being applied first and the nano-silica being applied and soaked in the second step.

Figure 7 shows the recording of the contact angle after hydrophobic coating and a demonstration of the self-cleaning function. Figure 7a shows the contact angle of 152.93° after the drop has fallen and stabilized in the contact angle test, which meets the superhydrophobic standard. Furthermore, Figure 7b demonstrates the process of the drop falling and then bouncing up and rolling on the hydrophobic surface, which, in practice, can also take away dust stains. The specific test results are shown in Figure 7c,d. Figure 7c shows two sets of glass samples with and without coating, respectively. Figure 7d shows the self-cleaning effect of the smart window coating combined with hydrophobicity. The same amount of soil debris was sprinkled on the same parts of them, and the correspond-

ing water droplet-cleaning effect was demonstrated after the rain. It is obvious that the addition of hydrophobic coating can better maintain the cleanliness of the glass surface, and the water droplet rolling down successfully carries the soil debris away from the glass surface. Figure 7e shows a physical image of the smart window glass and the visual effect of light transmission in reality. The coated surface also succeeded in maintaining the surface dryness, which greatly improved the visual effect and aesthetics and thus ensured light transmission.

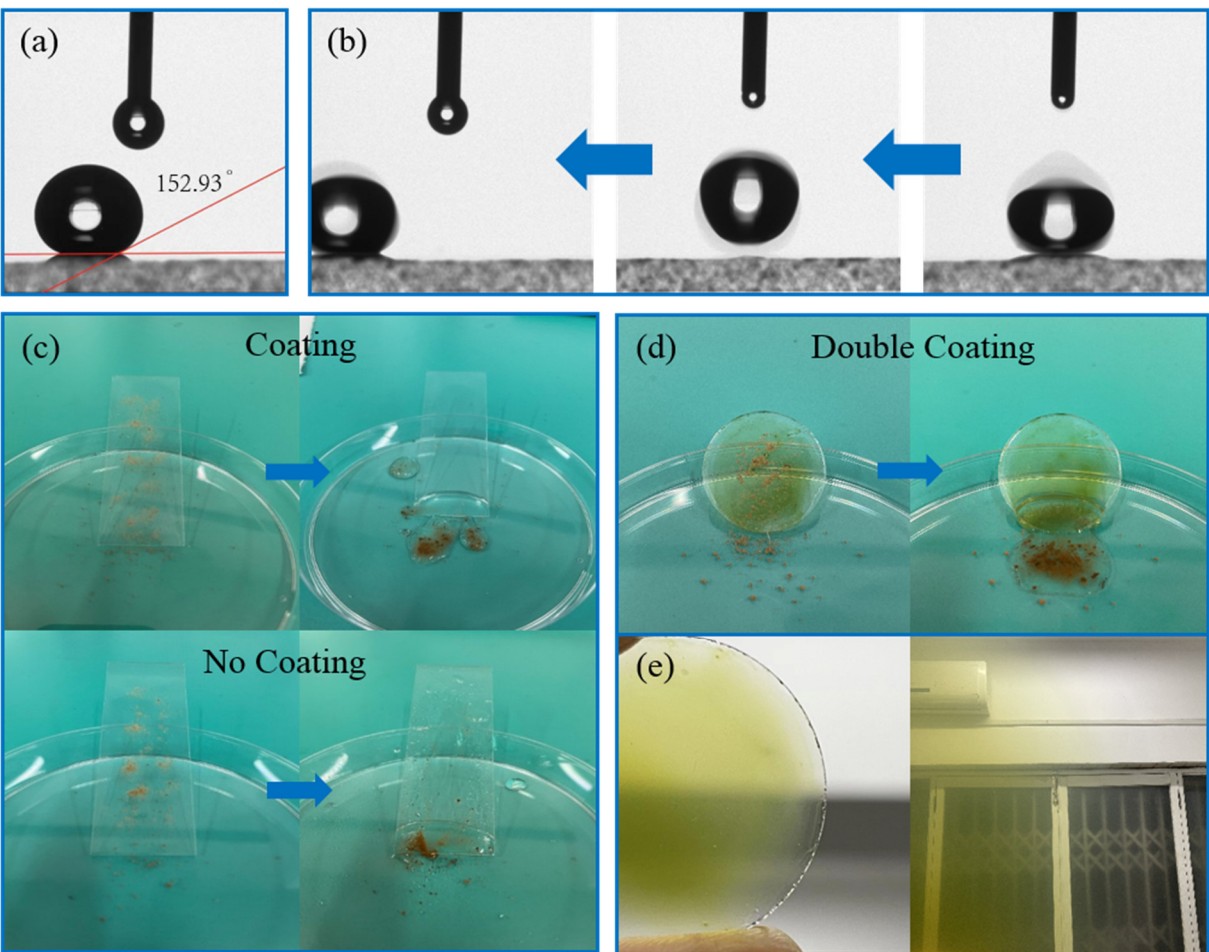

**Figure 7.** Hydrophobic and self-cleaning performance testing of smart windows. (**a**,**b**) The contact angle of smart windows and the droplet rolling process. (**c**,**d**) The schematic diagram of water separation and self-cleaning. (**e**) Actual viewing of the smart windows.

Table 3 shows the contact angle of 38.28° for the glass without the hydrophobic coating, 65.51° for the PVDF-HFP coating alone, and the change in contact angle after the coating is completed with the nano-silica in the two-step coating method.

**Table 3.** Influence of coating materials on surface wettability.

| Coating Materials | | Properties of Coated Fabrics | |
|---|---|---|---|
| **PVDF-HFP** | **Hydrophobic Nano Silica** | **CA (°)** | **Superhydrophobicity** |
| | | 38.28 | No |
| √ | | 65.51 | No |
| √ | √ | 152.93 | Yes |

## 4. Conclusions

In summary, $VO_2$ nanoparticles were successfully prepared using a simple hydrothermal recrystallization method. W-doped $VO_2$ nanoparticles were selected for performance testing and characterization of smart windows. XPS and XRD showed that W was successfully incorporated into the $VO_2$ (M) lattice to form a W-$VO_2$ solid solution. The doping of W successfully reduced the $VO_2$ heating phase transition temperature to 33.51 °C and 21.65 °C, which systematically improved the thermochromic performance of the material. Combined with the actual indoor and outdoor temperature difference, $VO_2$ with a doping amount of 1.7 at. % ($T_{c-heating}$ = 33.51 °C) is more appropriate for smart windows, the average luminous transmittance of the film reaches around 35% and the thermal modulation ability is 9.8%. It satisfies the functions of light and energy saving for smart windows. In terms of hydrophobic self-cleaning, the two-step coating method was used to transform the smart window glass to achieve a contact angle of 152.93°, which reaches the superhydrophobic condition, and the corresponding hydrophobic self-cleaning function was successfully performed in the soil chip self-cleaning experiment. This paper demonstrates that doping and hydrophobic modification can be successfully applied to smart windows. Furthermore, the manufacturing process is green and inexpensive, providing a theoretical and experimental basis for the promotion of energy-efficient building materials in the future.

**Author Contributions:** Conceptualization, X.S. and Z.X.; methodology, Z.X. and X.Y.; software, Z.X.; validation, Z.X., D.W. and H.Z.; formal analysis, X.S.; investigation, Z.X.; resources, T.Z.; data curation, Z.X.; writing-original, Z.X.; writing-review and editing, X.S., Z.X. and H.Z.; visualization, X.S. and H.Z.; supervision, J.Y.; project administration, Z.D.; funding acquisition, J.Z. All authors have read and agreed to the published version of the manuscript.

**Funding:** This research was supported by the Natural Science Foundation of Jiangsu Province (BK20180862, BK20190839) and the China Postdoctoral Science Foundation (2019M651725).

**Institutional Review Board Statement:** Not applicable.

**Informed Consent Statement:** Not applicable.

**Data Availability Statement:** Research data from this study will be made available upon request by contacting the authors.

**Conflicts of Interest:** The authors declare that this paper is completely original and free from any plagiarism or improper citation. This paper was carried out during the course of my study, research or work The authors declare no conflict of interest.

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
