# Peer review of "Application of W-Doped VO2 Phase Transition Mechanism and Improvement of Hydrophobic Self-Cleaning Properties to Smart Windows"

_photonics, doi:10.3390/photonics10111198_

Round 1

Reviewer 1 Report

Comments and Suggestions for Authors

VO2 has many advantages for the application of thermochromic smart window. This paper prepared W-doped VO2 nanoparticles by hydrothermal method, which allow the phase transition temperature was reduced to room temperature.This manuscript is well organized and contains enough information, and will be very interesting to those who working in this field. Thus, I think this paper can be accepted for publication after minor revision, and the following points should be addressed for the benefit of readers:
1. In the aspects of the preparation of VO2, some typical references should be added, such as "Journal of Materials Chemistry A, 2, 4520-4523","Small, 2017,13, 1701147", so that the readers could have a better and overall understanding of the synthesis of VO2 by hydrothermal method.

2. It is not clear about how to prepare the superhydrophobic coating on VO2 film. It is suggested to give a schematic illustration of the fabrication process of VO2 films and superhydrophobic coating.

3. In general, doping leads to a decrease in the luminous transmittance. It seems not this case in this paper. Furthermore, the information of the thickness of VO2 film was not provided, and it is better to present the photos of VO2 films with different thickness.

4. For thermochromic performance studies, it is seldom test the optical modulation separately in visible and infrared band, as shown in Fig. 5a and 5b. Please give a transmittance measurement range from 250 to 2500nm by the spectrophotometer such as the popular used UV-3600 equipment.

5. According to Fig. 5a, the average luminous transmittance of the film was around 30%, but the authors claimed the value is 59.6% in Fig.5d and in the Conclusion section. How to explain this inconformity? Please check the integral formula carefully.

6. In Fig. 6, there is no continuous VO2 film was observed. So, what is the reason?

7. Some grammatical errors should be double-checked, including the subscripts in VO2.

Comments on the Quality of English Language

Some grammatical errors should be double-checked, including the subscripts in VO2.

Reviewer 2 Report

Comments and Suggestions for Authors

The manuscript entitled Application of W-doped VO2 phase transition mechanism and improvement of hydrophobic self-cleaning properties to smart windowsinvestigates the solar modulation performance of smart windows with different W-doping and innovatively combines smart windows with hydrophobic work to enrich the usage scenarios of smart windows. I believe that this manuscript still has several issues that need to be improved by the authors as follows.

1.   Note the repeated use and expression of words and semantics. Typos, missing spaces, etc., must be carefully proofread.

2.   The subject of the paper should be This paper...'', not I.... The words used in the paper need to be more written, such as 'important' can be replaced with more formal words.

3.   The format of the main and subheadings in the paper should be standardized. Pay attention to the formatting of table annotations and check for problems such as centering of table headings.

4.   In Fig. 2, the XRD mapping is matched to this corresponding standard card for the major diffraction peaks, and there are distinct diffraction peaks that are not matched to the card that need to be explained in the article.

5.   In Eq. 2, the significance of (M) and (R) needs to be explained in the notes section.

Reviewer 3 Report

Comments and Suggestions for Authors

Dear authors,

your work is interesting for readers. Anyway, there is a critical question. With SEM analysis reported in Fig. 1  (where the reported scale is 3 micron) it is impossible to estimate the declared nanodimensions of "of tens to hundreds of nanometers". So, you have to add other SEM photos at different enlargement  for confirming this SUPPOSITION, or better a AFM, or TEM analysis. 

Best regards

Comments on the Quality of English Language

In Fig 6a, please remove the term Angel.
